# Artificial Water Troughs Use by the Mountain Ungulate *Ovis gmelini ophion* (Cyprus Mouflon) at Pafos Forest

**DOI:** 10.3390/ani12213060

**Published:** 2022-11-07

**Authors:** Nicolas-George Homer Eliades, Christos Astaras, Belle Verheggen Messios, Rob Vermeer, Kostas Nicolaou, Ilias Karmiris, Nicolaos Kassinis

**Affiliations:** 1Nature Conservation Unit, Frederick University, Pallouriotisa, Nicosia 1036, Cyprus; 2Forest Research Institute, ELGO-DIMITRA, Vassilika, 57006 Thessaloniki, Greece; 3Department of Animal Management, Van Hall Larenstein University of Applied Science, 8934 CJ Leeuwarden, The Netherlands; 4Game and Fauna Service, Ministry of Interior, Nicosia 1453, Cyprus

**Keywords:** wild mountain ungulates, water provision, water point, semi-arid regions, camera trapping

## Abstract

**Simple Summary:**

Human activities often affect the access and supply of water to wildlife, e.g., via water pumping, spring diversion, dam construction, and fencing. For this reason, wildlife managers have been provisioning surface water to wildlife in arid/semi-arid ecosystems for decades, typically via the construction of artificial water points. In this study, camera traps were used to examine for the first time the Cyprus mouflon’s (*Ovis gmelini ophion*) use of artificial water troughs in Pafos Forest—the stronghold of Cyprus’ threatened and endemic wild sheep. The ten monitored water troughs are part of the larger grid installed by the Game and Fauna Service to provide water to the mouflon during the prolonged dry season. Mouflon drinking was positively related to temperature, visiting the water troughs significantly more during late morning and midday hours of warmer days. There was no evidence of mouflon temporally avoiding water troughs used by predators (red foxes, feral dogs) or during hunting days. The findings suggest that water provision can be an important management tool for mediating, partially at least, the impact of global warming on water-dependent species such as the Cyprus mouflon, and therefore, additional studies focused on improving water trough effectiveness (e.g., regarding their spatiotemporal deployment) are advised.

**Abstract:**

For large herbivores inhabiting arid/semi-arid environments, water can be a limiting resource affecting their distribution and abundance for periods when water requirements are not met via forage. The Cyprus mouflon (*Ovis gmelini ophion*) is such a species, which is endemic to the mountain habitats of Cyprus. Recognizing water scarcity to be a major pressure to the mouflon, and with global warming projected to intensify hot and dry periods in the region, the Game and Fauna Service has been maintaining a network of locally designed watering troughs in Pafos Forest—the mouflon’s stronghold—since 1997. This study describes the mouflon’s use of the water troughs and examines whether visitation rates differed at the daily or weekly scale in response to environmental, climatic or anthropogenic parameters. Using camera traps, ten troughs were monitored from September 2017 to March 2018 (1,065 days; range 29–164 days per trough). Mouflon were detected at seven troughs (mean herd size 1.5 ± 1.2) during 373 independent detections (≥30 min interval between photographs), with visits peaking during late morning and midday hours. Generalized mixed-effect models showed mouflon visiting water troughs more frequently during hotter days, regardless of recent precipitation. Visits were also more frequent at water troughs located close to tar roads. Moreover, there was no evidence of mouflon avoiding water troughs used by predators (red foxes, feral dogs) at either daily or weekly scale, or during hunting days. The study supports the value of artificial water troughs for mediating, partially at least, the effects of climate change on mountain ungulates such as the Cyprus mouflon. Additional studies are proposed that will examine both mouflon drinking patterns across all seasons and ways of improving the effectiveness of the current water trough grid.

## 1. Introduction

Water is fundamental for all life on earth. For large herbivores, especially those inhabiting arid or semi-arid environments, access to surface water can be a critical resource that limits their distribution and abundance in periods when water requirements are not met via forage [1,2,3,4,5,6,7]. However, water is essential for human activities as well. Urban, agricultural, and industrial developments have reduced or degraded supply and wildlife access to natural sources of water, mainly via spring diversions, water pumping, dam constructions, and fencing [8,9,10]. In response, wildlife managers have been provisioning surface water in water-stressed ecosystems for decades, either via the construction of artificial watering points (and/or water troughs) or the modification of natural sources [5,11,12,13]. These water developments not only provide a critical resource during dry periods but they can also increase the available forage area of ungulates by providing surface water access near seasonally dry and therefore inaccessible areas. In doing so, artificial water troughs can be used to manage animal densities and support populations of rare species [3,13,14]. Moreover, they help mitigate the effects of anthropogenic barriers constraining wildlife access to surface water [7,15,16,17].

Reported side effects of artificial water troughs include impact on surrounding vegetation by increasing herbivore populations, altered animal movements, increased predation and disease transmission risk, and competition among species [4,7,18,19]. However, as human-induced climate change has already measurable effects on temperature and precipitation patterns in certain parts of the world, artificial water troughs gain broader recognition as a wildlife management tool [20,21,22,23], increasing at the same time the need to better understand their impact on wildlife.

According to climate models, the eastern Mediterranean region is a global climate-change hotspot, where mean annual temperature is expected to increase by as much as 4.1 °C within the 21st century [24,25]. Cyprus is a water-stressed island located in the eastern Mediterranean Basin, where—in addition to a temperature increase—a decrease in precipitation in the decades ahead is anticipated [25,26]. Under such extreme conditions of prolonged dry periods, thermoregulation will become even harder for water-dependent species such as the endemic and emblematic Cyprus mouflon (*Ovis gmelini ophion*) or “agrino”, as this wild sheep species is locally known. Although one of the smallest wild sheep (mean weight: male 31.6 kg/range 23–47 kg, female 22.4 kg/range 18–34 kg, [27]), the mouflon is the island’s largest terrestrial animal.

Once at the verge of extinction due to illegal hunting (<100 animals in 1937) [28], conservation efforts—including national (Cyprus Law 152(I)/2003) and international (Habitats Directive 92/43/EEC Annexes ΙΙ and IV; CITES Appendix I) protection status—have stabilized the Cyprus mouflon population by the end of the 20th century to more than 2000 (2574 ± 599) [27,29], according to the Game and Fauna Service (Ministry of the Interior, Republic of Cyprus) monitoring scheme introduced in 1997. The species is now ranked as Vulnerable according to the IUCN [30].

The species’ current distribution in Cyprus extends over 700 km^2^ in the Troodos Mountain range [27]. The main population occurs within the Pafos Forest, in which during the last decades, the species range expanded from the core forest area to the periphery, with the species density there currently being 2.5 times the density of its occurrence at the forest core [27,28]. In addition, in the last decades, the mouflon has expanded its range eastwards to the Troodos National Forest and to the south in mountainous areas outside forest boundaries [27,31].

Following a prolonged drought period from 1995 to 2000, the Water Department of Cyprus began boring new wells within the Pafos Forest, pumping the water to irrigate orchards. As a result, many springs located at the core of the Pafos Forest started drying up in the warmest months, while springs at the periphery did not [32]. While it is assumed that mouflon direct water consumption is low during most of the year [33], the species has been found to stay in the vicinity of water sites during hot and dry periods [34,35]. In cold and wet periods, green, succulent forage, the combination of lower temperatures, dew formation and precipitation may result in mouflon not requiring additional fresh water intake [36]. Dew can be particularly important, as in semi-arid environments—such as Cyprus—it can surpass precipitation in contribution to available water for wildlife [37,38]. Recognizing water scarcity to be a major pressure to the mouflon, at least seasonally, as it is a water-dependent ungulate with narrow distribution and an ecological niche limited to a mountain habitat, the Game and Fauna Service installed and maintaining a network of watering troughs in Pafos Forest from 1997 to 2020 (Figure 1) in order to provide water to the mouflons during the prolonged dry season, especially in areas with no surface water during this period. The water troughs were designed and constructed locally, specifically for the Cyprus mouflon, emphasizing low water evaporation, as they are supplied from water tanks (see Appendix A for design specifications). The troughs were placed predominately in areas lacking natural surface water in hot periods and in the driest areas of the forest, and they continue to be in operation to date. As the periphery of the forest borders agricultural areas, the water troughs were also meant to keep the mouflon within the protected area in order to reduce crop raiding and potential competition with livestock for forage, habitat and water [32,39].

The aim of this study is to provide a first description of the artificial water trough utilization by the Cyprus mouflon in Pafos Forest and to assess how it changes over space and time in response to environmental, climatic or anthropogenic parameters. Specifically, the current study investigated whether the mouflon visitation rate at ten water troughs, monitored using camera traps, differed at the daily or weekly scale in response to temperature and precipitation, topography, vegetation, distance to other water sources and the road network, use by potential predators of mouflon, livestock and hunting activity—the latter legally occurring at the periphery of the study area during certain weekdays. While limited in its duration (September 2017–March 2018) due to logistical limitations that precluded data collection across all seasons—including the warmest summer months, the study offers a first insight on the potential benefits and limitations of the current mouflon water provision strategy. Moreover, the study discusses additional, secondary benefits, of artificial water sites for the monitoring of protected species such as the mouflon, and it presents details of the locally designed water trough for use by managers at similar semi-arid landscapes.

## 2. Materials and Methods

### 2.1. Study Area

The study took place within the Pafos Forest (614 km^2^; Centroid: N 35.03°, E 32.67°), which is a state-owned forest located in the northwest part of the Troodos massif of Cyprus—the third largest Mediterranean island (9251 km^2^) (Figure 1). Extending from sea level to 1362 m (Mt Tripilos), the Pafos Forest is the island’s best-preserved forested area and the stronghold of the endemic Cyprus mouflon. The Pafos Forest consists of a core zone (263 km^2^) containing the peaks and higher slopes and a peripheral zone (351 km^2^) that extends to the lower elevation zones and borders agricultural areas. The forest is nowadays a part of the European Union’s Natura 2000 network of protected areas (SPA and SCI).

The climate condition in Pafos Forest is characterized, as the entire island, as extreme Mediterranean with mild and rainy winters from November to mid-March and prolonged, hot and dry summers from mid-May to mid-September. Transitions to autumn and spring are brief. The mean winter and summer temperature is 10 °C and 35 °C for lower altitudes and 5 °C and 22 °C at Mt Tripilos, respectively. Precipitation has also a pronounced altitudinal gradient, ranging from 300–350 mm to 1000 mm (mean: 473 mm for the 1991–2014 period) [40]. The topography is rugged with steep slopes and narrow streambeds, which are all dry except for the wet, winter season. Ten percent of the Pafos Forest’s 619 native plant species are endemic to Cyprus [40]. The forest core is dominated by *Pinus brutia* L. and the endemic *Quercus alnifolia* Poech, along with *Arbutus andrachne* L., *Pistacia terebinthus* L. and *Rhus coriaria* L. The forest edge is characterized by maquis vegetation with dominant species being the *Calicotome villosa* (Poir) Link, *Rhamnus oleoides* L., *Sarcopoterium spinosum* Spach and *Cistus creticus* subsp. *Creticus* [40,41]. Riparian vegetation grows at all elevations and consists of broadleaf species such as *Platanus orientalis* L., *Alnus orientalis* Decne., *Laurus nobilis* L. and *Myrtus communis* L. Trees are sparse at lower elevations, with vegetation being either phryganic or grasslands [40].

The fauna of Pafos Forest is also diverse, with 96 bird (five endemics) and 24 mammal species [41]—including the Cyprus mouflon. The social structure of the mouflon is similar to that of other wild sheep species; the same gender groups forage separately except for autumn, during the breeding (rut) season [42,43,44]. The Cyprus mouflon group size (mean 2.9 animals/group) is small compared to ungulates inhabiting more open habitats [27]. 

Although the mouflon population is legally protected and is long monitored through standardized autumn transects covering the species range and by lambing surveys calculating lamb to 100 ewe ratios [27], the species is still threatened. Based on autopsies of dead mouflon, major causes of mortality are disease (30%), predation by feral dogs or red foxes (25%), poaching (16%), and vehicle collisions (13%), with the rest being due to snake bites, accidents or undetermined causes [27]. Although the management plan for Cyprus mouflon prohibits its interaction with livestock, livestock grazing remains a potential threat to mouflon due to the risk of disease transmission and possible competition for forage, habitat and water resources [27,31,32]. 

Hunting for small game is allowed at the periphery of the Pafos Forest from the end of August to the end of February, but only on Wednesdays and Sundays, whereas parts of the NE Pafos Forest are allowed for hunting from November to December with hare (*Lepus europaeus cyprius*) being the primary quarry. Dogs are used during hare hunting, and it is in part due to their presence that mouflon vigilance increases during the hunting season [27,31,32].

### 2.2. Field Work Design

Camera traps were used to monitor the use of ten artificial water troughs by wildlife, as this method has been used extensively in similar studies across the world [7,13,45,46]. The monitored water troughs were located at the forest core (*n* = 3), where mouflon density is usually low, and the higher density periphery zone (*n* = 7) (Figure 1). The selection of water troughs to monitor was based on the estimated mouflon density from transect count estimates (selecting sites of varying densities—low (*n* = 5; 1–5 mouflon/km^2^) to high (*n* = 5; 5–10 mouflon/km^2^)) and the presence of vegetation near the water troughs where the camera could be camouflaged in order to reduce risk of theft by people.

Differences in monitoring duration and timing among water troughs were due to logistical limitations and occasional camera malfunctions (Table 1). Most water troughs were recorded for the November 2017–March 2018 period, while two were monitored for the September–October 2017 period. Three different camera trap models (seven Ltl Acorn 6210 MG, Desa Moineis, USA; two Reconyx HC600 Hyperfire H.O. Covert IR, Holmen, USA; one Bushnell Trophy Cam119466, Kansas, USA) were used in the current study, all of which were programmed to record photos and videos when triggered by movement, with a 1 sec interval between three photos, which was followed by a 10 sec interval for video (duration of videos 6–8 s), after which the camera could be retriggered by movement of wildlife. All cameras were able to capture night images using infrared flashes, and we recorded the date and time of each image and video. The cameras were placed at approximately 40 cm from the ground and at 3–5 m from the water troughs, aiming at them, and therefore—unlike with camera set ups monitoring wildlife along a trail—had ample time to “wake up” when movement was detected and record animals that stopped to drink water. While the impact of the camera model on the detection probability of wildlife has been previously reported [7,19,47,48], preliminary analysis showed that camera models were not a significant predictor of mouflon detections at either the 6 hr period or the weekly scale. The single camera set up at each water trough was able to document effectively animals approaching the artificial water troughs to drink (see Appendix B). The camera traps were visited every two weeks to replace batteries and SD cards. A more frequent data collection protocol was avoided to reduce the previously reported effect of human presence on the site’s use by mouflon [27,28,36,49].

In addition, a camera trap was used to record wildlife presence at each of five known mouflon forage areas (e.g., openings cultivated and sown by Game and Fauna to provide food for mouflon)—two within the core zone and three at the periphery of Pafos Forest. These cameras were deployed only during the winter to early spring months (Table 1), and the data were used to determine mouflon forage activity patterns, in order to compare them to concurrent activity at the artificial water troughs.

### 2.3. Data Analyses

The data from the camera trap SD cards were transferred to a computer, where they were reviewed independently by two researchers (BVM, RV). For each photo/video, the species, date, time, and site were recorded. Photos/videos of the same species with up to 30 min intervals were considered as part of the same independent detection (henceforth referred to as detection) [45]. Only the time of the first photo/video was used in the analysis from a series of records. All animals observed next to the artificial water troughs were assumed to be there to drink water, regardless if they were actually drinking in the photos/videos or not, as there was not continuous video to establish with certainty if animals were drinking. Nevertheless, the mean visit duration was >4 min, suggesting that the mouflon were visiting the trough rather than just passing by. In the case of mouflon, the minimum herd size (i.e., max number of individuals recorded in a single photo/video) and the estimated age (adult, sub-adult, juvenile) and sex of observed individuals were recorded, even though it was not assumed that all animals of a herd were observed.

To assess whether mouflon visitation rates at the artificial water troughs varied in time or space due to environmental, climatic or anthropogenic parameters, the mouflon independent detections (each detection counting as one, regardless of the number of individuals observed) were first clustered at two temporal scales; at the 6 hr scale (*n* = 4859 detections per 6 hr period; early morning 00:00–6:00, late morning 6:00–12:00, midday 12:00–18:00, and evening 18:00–00:00) and the week scale (*n* = 144 weekly detections; week defined as Monday to Sunday). Since the mouflon visits were not normally distributed (Shapiro–Wilk normality test, *p* < 0.001) at either of the temporal scales considered due to the presence of a lot of periods with zero visits, a series of negative binomial generalized mixed effect models were constructed for each time scale in R package lme4::glmer.nb [50]. The site (trough) and day were included as nested random effects (fixed slope, random intercept) for the 6 hr models, and site and week were included for the week models. Using the R package DHARMa:testZeroInflation [51], it was determined that the use of zero-inflated negative binomial models was not required.

Both series of models considered as site (trough) the following variables: (i) elevation, (ii) southern exposure, (iii) distances to nearest: (a) road, (b) terrace/cliff (since mountain ungulates use rocky areas to escape predators [52], (c) sown forage patch, and (d) water trough, (iv) mean distance to the three closest water troughs, (v) number of artificial water troughs within 2 km radius, (vi) percent of open (non-forest) area within 2 km radius, (vii) mouflon density (mouflon per km^2^), (viii) predator encounter rate, and (ix) location (core or periphery of Pafos Forest) (see Appendix C for detailed description of variables). The 2 km radius used to estimate certain parameters was based on the mean daily travel distance of male mouflons (2 km; [31]). Site level parameters were calculated using GIS software (QGIS 3.16.14) (https://qgis.org/en/site/; accessed on 1 August 2021). Time-related variables were precipitation (day and previous week’s total), temperature (mean, max, min), and—only at the 6 hr scale models—the time of the day (TOD; i.e., early morning, late morning, midday, evening) and whether it was a hunting day (see Appendix C). Temperature and precipitation data were obtained from the closest weather station (Kampos), which is located within the Pafos Forest (range 5–15 km from the monitored water troughs).

The Akaike Information Criteria (AIC) was used for model selection [53]. First, univariate models were run and compared against the null (intercept only) model, including in multivariate models only informative variables (i.e., their univariate model had <AIC than the intercept only model). This way, an optimal set of informative variables was identified while also managing model complexity. In the case of correlated variables (r > |0.7|), only the one with the lowest univariate model AIC was kept. Variables not in the 0–1 scale were standardized (z-score) except for distances, which were included as log values in the models. Once a final set of fixed variables was selected, all possible multivariate combinations were run using MuMIn:dredge [54]. The informal diagnostics in R package DHARMa were used to review the distribution and structure of the final model’s residuals. The goodness-of-fit of the best models (one for each time scale considered) was assessed using the coefficient of determination (R^2^) [55], which calculates the correlation between fitted and observed values.

The R package “overlap” [56] was used to estimate the coefficient of activity overlap (Δ), as defined by [57], between: (a) mouflon at water troughs and forage sites, (b) mouflon at water troughs during hunting and non-hunting days, and (c) mouflon and foxes at water troughs. A Δ value of 0 denotes no activity overlap across 24 h, and a value of 1 denotes a perfect overlap. Bootstrapping (*n* = 1000) was used to estimate the confidence intervals of Δ.

## 3. Results

The total camera trapping survey effort at the ten monitored water troughs was 1025 days. Data from 96.5% of those days (985 days; 98.9 ± 39.2 SD days/camera trap; range 29–164 days) could be used, and for the rest, either vegetation obstructed the camera’s field of view or the camera did not temporarily record due to water ingress. In total, 27,642 photos were recorded. Fourteen species of wildlife were observed in 987 photographs: ten bird species (*n* = 363 photos) and four mammal species (*n* = 624 photos) (Table 2). No livestock were recorded at any of the water troughs, and all dogs were stray (feral). 

Mouflon were detected at seven of the water troughs, accounting for 88.8% of mammal photos (*n* = 555). Of the three water troughs with no mouflon detections, two were in low mouflon density areas at the core of the Pafos Forest (sites DR, SA), and one was at the water trough with the highest mouflon density located at the periphery of the protected area (site DF). The duration of the mouflons’ visits were 4.6 ± 4.9 min (range of all visits 0.3–27.5 min). In total, there were 373 independent detections (i.e., at least a 30 min interval between consecutive photographs), which formed the data for the current analysis. The mean mouflon detections per day per camera trap was 0.4 ± 0.62 SD (range 0–2.02) (Table 3).

The number of mouflons observed in a detection ranged from 1 to 10 animals. The total number of mouflon detected was 595 (based on the minimum number of animals that could account for the sex and age of the mouflon observed in all photos of that detection; since herds probably repeatedly visited the water troughs and it was impossible to differentiate between them, this should not be viewed as a population estimate of the mouflon using the monitored troughs). Solitary animals accounted for 75.8% (*n* = 283) of all detections and 100% of all December to March detections (Table 4). Adult males were the age and sex group most likely to be observed alone (75.6%, *n* = 214 detections). Adult females were mostly seen in groups (64% of 103 adult female detections). Young females were always observed in groups, while solitary young males were observed only at one water trough (KK) during a 15-day period of November (*n* = 11). Solitary juveniles were observed only once. Mean group size was 1.5 ± 1.2 mouflons (3.3 ± 1.4 excluding solitary detections). The mean group size was significantly higher during late morning periods (*p* < 0.001; early morning: 1.2 ± 0.6, late morning: 1.9 ± 1.6, midday: 1.4 ± 1, evening: 1.4 ± 1.1). At the level of troughs, mean group size was significantly higher at Artratsa (AR, 2.0 ± 1.7; *p* = 0.008) and significantly lower at Kokinokremmos (KK, 1.2 ± 0.7; *p* = 0.024) (Table 5). 

Of the group detections (*n* = 90), 30 involved single-sex adult mouflon groups (adult males: *n* = 13, range across all sites 2–4 animals; adult females: *n* = 17, range across all sites 2–5 animals). In the mixed sex groups (*n* = 60), adult males were present in most detections (*n* = 46, mean 1.7, range across all sites 1–4 animals), as were adult females (*n* = 49, mean 1.8, range across all sites 1–8 animals).

Overall, of the 595 mouflon detected, adult males accounted for 56.1% of the animals, adult females 29.9%, young males 4.7%, young females 0.7%, juveniles 3.3%, and unknown age/sex 5.3%. Excluding solitary animal detections, adult females were the most abundant demographic age/sex group, accounting for 46.1% of the 295 mouflon detected in groups (≥2 mouflon). Adult males were second (37.3%), followed by young males (5.4%), unknown (3.7%), juveniles (6.1%), and young females (1.3%).

At the 6 h temporal scale, the mouflon visitation rate at the water troughs was best explained by four competing models (AIC < 2), with the following environmental and climatic variables having the highest explanatory values: distance to roads, distance to terrace/cliffs, time of the day, maximum temperature (24 h), and precipitation (24 h and 7-days total). The model average coefficients of these models are presented in Table 6. The Dharma non-parametric dispersion test of the standard deviation of fitted vs. simulated residuals did not show significant dispersion (*p* = 0.16). The coefficient of determination of the global model (R^2^ = 0.562) suggests that these variables explain well the observed variation in mouflon visitation rates.

According to the modeling results, mouflons were significantly more likely to visit the water troughs located closer to tar (asphalt) roads, especially during late morning and midday hours of warmer days. Evening visits were less likely but still significantly more frequent than early morning ones (intercept). Distance to terrace/cliffs and 24 h precipitation had a negative but not significant effect on mouflon visitation rate at this temporal scale. Adding to the global model information on whether hunting was allowed or not on a given day did not improve the model (Likelihood Ratio Test, *p* = 0.571). 

At the week temporal scale, the mouflon visitation rate at the water troughs was best explained by six competing models (AIC < 2) that included distance to roads, distance to terrace/cliffs, mean maximum temperature (week), predator (fox and dog) encounter rate and a site’s southern exposure as predictive variables for the utilization of water troughs by the Cyprus mouflon. Table 7 presents the model average beta coefficients. The Dharma non-parametric dispersion test of the standard deviation of fitted vs. simulated residuals did not show significant dispersion (*p* = 0.21). The coefficient of determination of the global model (R^2^ = 0.979) shows that the average model explains very well the observed variation in mouflon visitation rates at the week temporal scale.

Specifically, as with the 6 hr scale analysis, mouflons were more likely to visit water troughs located close to tar (asphalt) roads during warmer days. Distance to terrace/cliffs was again negatively related to mouflon visitation but still not significantly so. Predator encounter rate at a water trough did not deter mouflon from visiting; in fact there is a significant positive relation at the 0.1 level. Similarly, water troughs with higher southern exposure were more likely to be visited, but the significance is again at the 0.1 level. Adding hunting day information did not improve the model (Likelihood Ratio Test, *p* = 0.491).

To examine whether the absence of mouflon detections at the Den of Foxes (DF) water trough masked the overall effect of mouflon density on water trough visitation rates (since it is located in the highest mouflon density zone but also close to a den of a predator), the above analyses were run—at both temporal scales—excluding the DF data. There were no differences in the list of significantly informative variables at the univariate level, including the mouflon density.

The mouflon had a primarily diurnal and largely overlapping (Δ = 0.64; 0.51–0.79 95% CI) activity pattern at the water troughs and forage sites (Figure 2). At water troughs, mouflon showed a bimodal pattern, with visits peaking after dawn and before dusk. Foraging peaked after the morning watering period and declined sharply following the evening watering period. There was no significant difference in the activity of mouflon groups consisting of females with young and that of other groups (Δ = 0.72; 0.54–0.86 95% CI) (Figure 3). Mouflon activity at the water troughs during days with and without hunting was similar (Δ = 0.91; 0.84–0.97 95% CI) (Figure 4). The mouflon and red fox activity overlap at water troughs was high (Δ = 0.6; 0.49–0.7 95% CI), although, overall, foxes visited the water troughs earlier in the morning and later in the evening than the mouflon (Figure 5). 

## 4. Discussion

The current study describes for the first time the utilization of artificial watering points (water troughs) by the Cyprus mouflon (*Ovis gmelini ophion*) in Pafos Forest. Despite the limited spatial and temporal scale of the study, it presents evidence that the troughs are important for the water provision of the species, particularly during autumn, which along with summer constitute the dry season in Cyprus and the onset of mouflon mating (rut) season—a period when rutting animals require extra energy and water [58]. Specifically, the sharp decline in mouflon detections at the water troughs from December onwards—once the rains start—suggests that mouflon presence at the water troughs was not coincidental as a result of random movement in the landscape during foraging, but it is purposeful. Mouflon water drinking activity is bimodal, with the peak after dawn being more pronounced than the one before dusk, which is a pattern that has been reported for some ungulates of semi-arid regions (e.g., bighorn sheep, [59]; kudu/roan antelopes, [60]; impala, [7]; mule deer/pronghorn [5]). 

Since the watering requirements of wild ungulates can be governed by a series of factors, including individuals’ physiology (e.g., thermoregulation needs, reproduction cycle, physical condition [12,59]) and climatic variables (e.g., ambient humidity, precipitation, temperature [5,12,61]) temporal variations in mouflon visitation rates at the water troughs were anticipated. The reported significantly higher visitation of water troughs in warmer days and weeks can be explained by the increased thermoregulation needs of ungulates at these periods, as water is needed to lower body temperature via evaporation [12,62]. High temperatures also reduce available water for intake via forage [61,63,64]. Precipitation, while significantly informative when included as the sole explanatory variable of water trough visitation at the 6 h scale, was not significant at the final multivariate model (during the study period). This was true for both daily precipitation and total precipitation over the past week. Precipitation therefore, at least for the seasons studied, is not readily available to the mouflon, or if it is, it is not sufficient to alter the use of the water troughs.

The reported reduction in mouflon watering and foraging activity in early afternoon (Figure 2) is consistent with reports that Cyprus mouflon seek shade at that time of the day [36]. Moreover, a similar bimodal activity has been reported for the European mouflon (*O. gmelini musimon*) in France [65] and Croatia [66], and—albeit more crepuscular than diurnal—for the Sardinian mouflon (*O. orientalis musimon*) [67]. It is, however, at odds with an earlier assumption that the Cyprus mouflon drank water at night, which was inferred from night observations of mouflon foraging at night in Pafos Forest [28,49]. The findings show that mouflons will utilize water resources available to them during the hotter periods of the day (e.g., water needed for thermoregulation). Another explanation could be that the mouflon are less wary of human presence, as human presence itself has become less dense (e.g., illegal logging and grazing within the study site). Activity pattern changes across sites or time in response to human and predator activities is well documented for mammals, including ungulates [68,69,70]. The lack of water trough visitation differences among hunting and non-hunting days reported here suggests that at least at some spatial and temporal scale, mouflon are more tolerant of human presence than reported in older studies. More studies are needed to determine whether the activity pattern observed during this study persists year round (and especially during July–August, the hottest months of the year) and across the Pafos Forest. Studying mouflon activity in areas without water troughs, if this opportunity arises, would be especially informative. Numerous factors have been reported to influence a species’ choice of watering site, such as distance to forage, availability of alternative water sources [71], water quality including mineral content [72], surrounding vegetation (e.g., for cover and shade; [71,73]), interspecific and intraspecific competition [74], and occurrence of predators [75]. With the exception of distance to roads, no other spatial variable was significantly informative for mouflon visitation rates at the ten monitored water troughs. While mouflon density did vary across the study area, it was not an informative variable in either of the temporal scales examined (i.e., 6 h and weekly). This suggests that the spatial distribution and density of the water troughs deployed by the Game and Fauna Service are sufficient to keep competition for access to them low. 

Distance to tar (asphalt) road was the sole spatial parameter to be significantly informative (positively related) to mouflon water trough visitation, and this was true at both time scales examined. Tar roads are areas where the probability of encountering humans is higher, so the finding is surprising, since personal observations by the authors (N.K., K.N., R.V., B.V.M.) show a species that is intolerant of proximity to humans, and animals of all age will invariably flee when humans are detected at distances <200 m approximately. Regardless, even if mouflon were tolerant of potential human encounters, it does not explain the apparent preference for water troughs located near roads. Considering that vehicle collision is one of the major causes of mortality for the species, it is important that future studies examine the underlying drivers of the reported mouflon preference for water troughs closer to tar roads, so as to suggest—if necessary—adjustments to the water trough grid design. Mouflons appear to prefer water troughs located closer to terraces and cliffs, where they can retreat to safety when threatened. However, this relation was not significant at the multivariate level. It was persistent, though, at both temporal scales considered (6 h and weekly). Similar patterns have been reported for artificial water point and forage site use of other mountain ungulates, which like the mouflon use rugged, rocky areas to escape predators, such as the Desert Bighorn sheep *Ovis canadensis mexicana* [14] and the Nubian ibex *Capra nubiana* [76]. At the weekly scale, mouflons also visited more often water troughs located in areas with higher southern exposure (percent of area within 1 km radius facing SE to SW), but this observation was not significant (*p* = 0.1). South-facing slopes in the northern hemisphere are more exposed to solar radiation [77], and therefore, the thermoregulation needs of mouflon foraging in such areas will be higher compared to areas with low southern exposure. The above two findings suggest that a possible relation of water trough use and its location in terms of southern exposure and vicinity to terraces/cliffs should be examined more closely in the future, as it could provide guidance on where additional water troughs could be placed. 

Placement of the water troughs, current and future, should also take into consideration their possible impact on surrounding vegetation. Although ungulate aggregations near natural water sources do not appear to have a negative impact on adjacent plant communities [64], there are several studies that reported vegetation changes near artificial water sources due to trampling and overgrazing, especially when placed in sensitive habitat types [17,78]. One of the benefits of the water troughs designed and used by the Game and Fauna Service is that they are mobile. The actual water trough weighs <10 kg and the water is provided by a 700–1000 L plastic water tank, which—when empty—is relatively easy to lift and transport. A study should examine if and how water troughs impact the surrounding vegetation structure and community, and how long it takes for the vegetation to recover once a water trough is removed. This information is essential for determining whether the water troughs should be periodically relocated or deactivated and if so how frequently.

Another widely recognized concern of water provisioning is that it may increase the densities of prey species around the artificial water sources and therefore attract predators [5]. This increased risk of predation may be unacceptable for species of conservation concern. Red foxes, which together with feral dogs are the mouflon’s primary predators and account for a quarter of mouflon mortality [27,79], visited the water troughs mostly at dusk and dawn. This is expected given the species’ crepuscular activity [80]. The study’s findings do not show mouflons avoiding foxes at the water troughs at either of the temporal scales examined. One possible explanation is that, at least for the seasons monitored, mouflons do not consider the risk of fox predation to be high, since the lambs are >5 months old; therefore, due to similar behavioral and physiological needs, both species use the water troughs on hot days. Another explanation, though, could be that foxes are attracted to the water troughs due to their use by prey species. Predator activity pattern changes near artificial watering sites in semi-arid areas have been reported [5,12,81]. In addition, at the water trough “Den of Foxes” (DF), no mouflon were observed for the entire study period even though the area has one of the highest mouflon densities, which could be due to its vicinity to the fox den. Moreover, after November, only solitary mouflon were observed at the water troughs. While this may be primarily due to mouflon needing less water for thermoregulation in cooler weather, late winter/spring is also the time that females are pregnant, which is why they seek shelter in areas inaccessible to predators (e.g., cliffs, terraces, thick vegetation) where they are known to give birth [82]. Predation risk at water troughs should be examined further, ideally by studying the spatial ecology of both mouflons and red foxes in higher spatial and temporal resolution (e.g., with GPS telemetry collars). It should be noted, however, that the low water capacity of the troughs (3 lt) and the deployment of a large number of them in Pafos Forest is probably not conducive to supporting large mouflon densities around each trough or for predators to choose one to ambush wildlife. 

Intraspecies and interspecies disease transmission has been identified as a potential risk at artificial watering sites [83,84]. Considering that disease is responsible for one-third of all known mouflon mortalities [27], the artificial water troughs should be periodically cleaned [85]. In fact, a recent study examining Cyprus mouflon endoparasites in fecal samples (*n* = 104; incl. from the study area) [86] found parasites (mostly lungworms) in 97.39% of the samples. While no livestock were observed at the ten monitored water troughs during the study, disease transmission is well documented at the wildlife–livestock–human interface [87], so this possibility should be examined in areas where pastoralists bring their herds at grazing areas particularly at the north to northeastern and south site of the Pafos Forest (where illegal grazing is often recorded within the forest boundaries). While the water trough grid was developed by the Game and Fauna Service to increase the range available to the Cyprus mouflon during the driest periods of the year and to help mitigate the effects of anthropogenic barriers to surface water access for Pafos Forest wildlife, the study’s findings also show that is a valuable tool for monitoring the mouflon (e.g., population trends, herd structure, physiological condition, disease outbreaks). For instance, the findings on mouflon herd composition (demography) and mean group size presented helps the Pafos Forest management keep track of the population and ethology of the species in an efficient (low cost) and non-invasive way.

## 5. Conclusions

According to the International Panel on Climate Change, the eastern Mediterranean region is one of the world’s most vulnerable regions to the impact of global warming (IPCC 2013; IPCC 2018). In Cyprus, it is projected that by the mid-21st century, annual rainfall will decrease by 2–8%, maximum temperature will increase by 2 °C in the summer, and each year, there will be two more weeks of very hot days as well as longer consecutive dry periods by nine days [26,88]. Under these weather conditions, thermoregulation will become even harder for water-dependent species, including the Cyprus mouflon. 

The findings presented here suggest that manipulating surface water availability via a grid of artificial water troughs may be an important management tool for mediating, partially at least, the effects of climate change on the mouflon. The results also serve as baseline data for future research on the effects of water provisioning on wildlife in Pafos Forest and beyond. Simply documenting the mouflon’s use of water troughs, though, is not sufficient to understand their value as a wildlife management tool. It is important to study water trough use across all seasons and to understand better the physiology and activity patterns of mouflon during water-stress periods. Moreover, since the heterogeneity of surface water availability in space and time is important for ecosystem resilience and biodiversity [4], studies should examine the direct and indirect effect of the current water trough grid on Pafos Forest’s flora and fauna, in order to adjust its design, if needed, according to the management authority’s conservation priorities. Such studies could include, but not be limited to, spatial ecology and activity studies of mouflon using GPS telemetry collars, mouflon and other wildlife occupancy, and habitat selection in reference to seasonal surface water availability using camera traps, assessment of vegetation community changes over time in areas adjacent to water troughs, and potential interactions with livestock. With such additional information, the water trough grid could be amended by for instance moving the water troughs to new sites periodically or disabling them at times of high natural water availability.

## Figures and Tables

**Figure 1 animals-12-03060-f001:**
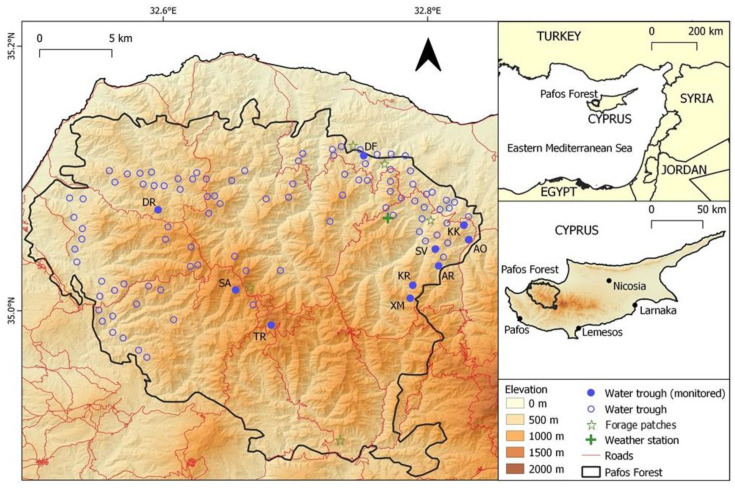
Map of the Pafos Forest area where the network of water thoughts was established for the watering of Cyprus mouflon. (DR: Dromonas; SA: Selladi Arnion; TR: Tripilos; DF: Den of Fox; SV: Selladi Voullas; KK: Kokinokremmos; AO: Apo Orkontas; AR: Artratsa; XM: Xeromoutas; KR: Koriftos).

**Figure 2 animals-12-03060-f002:**
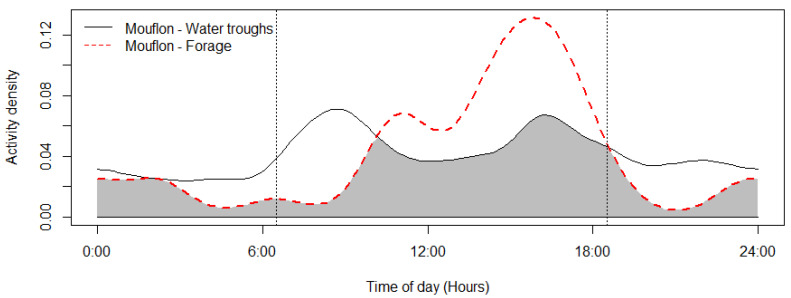
Mouflon activity overlap at monitored water troughs (*n* = 10) and forage sites (*n* = 5) (coefficient of overlap Δ = 0.64). Sunrise and sunset are marked at their average time for the study period (gray zone shows the overlap area).

**Figure 3 animals-12-03060-f003:**
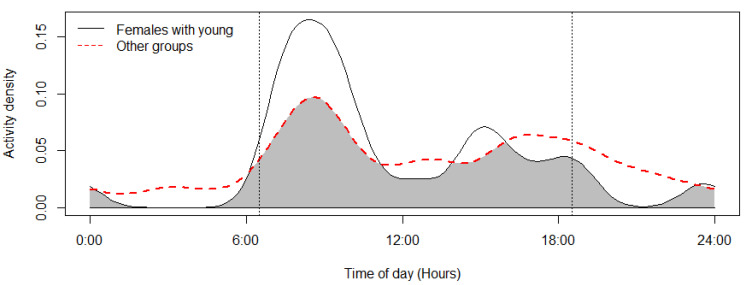
Activity overlap at the monitored water troughs of mouflon groups with females and young and other mouflon groups (coefficient of overlap Δ = 0.72). Sunrise and sunset are marked at their average time for the study period (gray zone shows the overlap area).

**Figure 4 animals-12-03060-f004:**
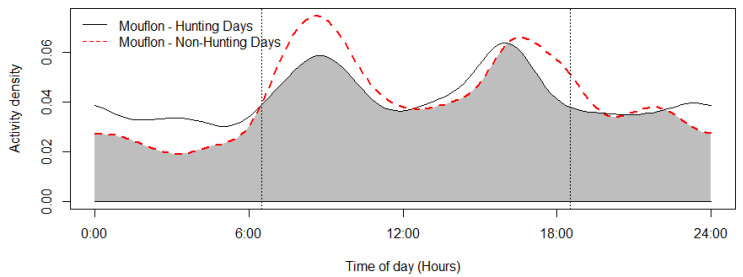
Mouflon activity overlap of days with and without hunting (coefficient of overlap Δ = 0.91). Sunrise and sunset are marked at their average time for the study period (gray zone shows the overlap area).

**Figure 5 animals-12-03060-f005:**
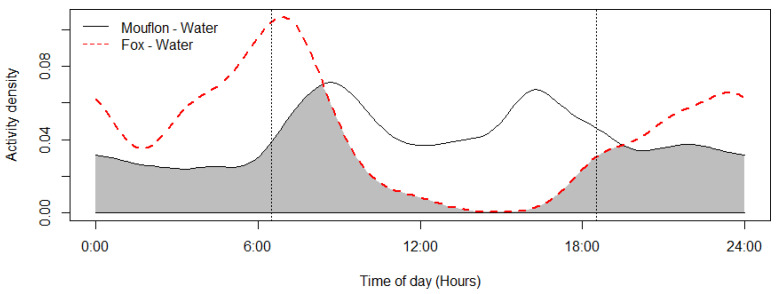
Mouflon and red fox activity overlap at the ten monitored water troughs (coefficient of overlap Δ = 0.6). Sunrise and sunset are marked at their average time for the study period (gray zone shows the overlap area).

**Table 1 animals-12-03060-t001:** Camera trapping survey period and effort per monitored site.

Site Name	Site Code	Location	Survey Period	Survey Effort (Days)
Apo Orkontas	AO	Water trough	26 September 2017—9 March 2018	164
Artratsa	AR	Water trough	2 September 2017–16 October 2017	44
Den of Foxes	DF	Water trough	21 November 2017–9 March 2018	108
Dromonas	DR	Water trough	16 November 2017–15 December 2017	29
Kokinokremmos	KK	Water trough	31 October 2017–9 March 2018	129
Koriftos	KR	Water trough	1 November 2017–9 March 2018	128
Selladi Arnion	SA	Water trough	18 December 2017–9 March 2018	81
Selladi Voullas	SV	Water trough	1 November 2017–9 March 2018	128
Tripilos	TR	Water trough	23 December 2017–9 March 2018	76
Xeromoutas	XM	Water trough	30 October 2017–9 March 2018	130
Barisia	VA	Forage area	20 January 2018–9 March 2018	48
Kotziakari	KZ	Forage area	31 January 2018–21 February 2018	21
Potamos Kampou	PK	Forage area	18 December 2017–9 March 2018	81
Psilo	PS	Forage area	18 December 2017–31 January 2018	44
Xeros	XS	Forage area	21 February 2018–9 March 2018	16

**Table 2 animals-12-03060-t002:** Number of camera trap photographs recorded per water trough site and species (dotted line separates birds from mammals).

Species/Site Code *	AO	AR	DF	DR	KK	KR	SA	SV	TR	XM	Total
Blackbird (*Turdus merula*)									5		**5**
Chukar partridge (*Alectoris chukar*)	5	2	1			1				1	**10**
Coal tit (*Periparus ater*)		1		2						1	**4**
Common kestrel (*Falco tinnunculus*)					1						**1**
Common woodpigeon (*Columba palumbus*)	107			34	4	10	1	76	29		**261**
Eurasian jay (*Garrulus glandarius*)	1			14			14	16	8		**53**
European robin (*Erithacus rubecula*)					1					1	**2**
Greenfinch (*Chloris chloris*)		3					3	3			**9**
Hawfinch (*Coccothraustes coccothraustes*)							1				**1**
Song thrush (*Turdus philomelos*)		1			1			3			**5**
Unidentified bird species	1	1			2			6	1	1	**12**
Brown hare (*Lepus europaeus cyprius*)		3	1		2					1	**7**
Cyprus mouflon (*Ovis gmelini ophion*)	226	163			114	21		21	7	3	**555**
Red fox (*Vulpes vulpes*)	1	3	19	2	6	11		4	2		**48**
Dog (*Canis familiaris*)	1				10			2		1	**14**

* Abbreviations of site codes: DR: Dromonas; SA: Selladi Arnion; TR: Tripilos; DF: Den of Fox; SV:Selladi Voullas; KK:Kokinokremmos; AO:Apo Orkontas; AR: Artratsa; XM: Xeromoutas; KR: Koriftos.

**Table 3 animals-12-03060-t003:** Monthly and overall (study duration) independent mouflon detections per survey day for each monitored water trough (dashes denote months when water troughs were not monitored). Monthly daily mean temperature and total monthly rainfall during the study period are also provided.

Site Code *	Detections/Day							Total Events (Overall Detections/Day)
	Sept.	Oct.	Nov.	Dec.	Jan.	Feb.	Mar.	
AO	0.00	0.74	4.27	0.03	0.00	0.00	0.00	152 (0.93)
AR	3.07	0.00	-	-	-	-	-	89 (2.02)
DF	-	-	0.00	0.00	0.00	0.00	0.00	0 (0.00)
DR	-		0.00	0.00	-	-	-	0 (0.00)
KK	-	3.00	2.7	0.1	0.00	0.00	0.00	87 (0.67)
KR	-	-	0.53	0.00	0.00	0.00	0.00	16 (0.13)
SA	-	-	-	0.00	0.00	0.00	0.00	0 (0.00)
SV	-	-	0.47	0.1	0.00	0.04	0.00	18 (0.14)
TR	-	-	-	0.00	0.19	0.00	0.00	6 (0.08)
XM	-	0.00	0.03	0.06	0.00	0.00	0.00	3 (0.03)
**Total events**	**91**	**26**	**240**	**9**	**6**	**1**	**0**	**373 (mean 0.4)**
Temperature (°C)	25.5	22	17.7	15.1	13.5	14.8	15.8	-
Rainfall (mm)	0	10.3	2.2	37.1	99.3	68.5	0	-

* Abbreviations of site codes: DR: Dromonas; SA: Selladi Arnion; TR: Tripilos; DF: Den of Fox; SV: Selladi Voullas; KK: Kokinokremmos; AO: Apo Orkontas; AR: Artratsa; XM: Xeromoutas; KR: Koriftos.

**Table 4 animals-12-03060-t004:** Number of solitary and group mouflon events (independent) and mean group size per month.

Month	Independent Detections	% Solitary	Solitary Events * (*n*)					Group Size (All)	Group Size (excl. Solitary)
			AM	AF	YM	J	Unk		
September	91	63.7%	40	16	-	-	2	2 ± 1.71	3.76 ± 1.79
October	26	42.3%	8	1	-	1	1	2.31 ± 1.32	3.27 ± 0.93
November	240	82.5%	151	19	11	-	17	1.33 ± 0.86	2.9 ± 1.13
December	9	100%	9	-	-	-	-	1	-
January	6	100%	6	-	-	-	-	1	-
February	1	100%	-	1	-	-	-	1	-
March	-	-	-	-	-	-	-	-	-

* AM = Adult Males, AF = Adult Females, YM = Young Males, J = Juveniles, Unk = Unknown.

**Table 5 animals-12-03060-t005:** Mean group size of mouflons per monitored water trough (includes solitary animal detections).

Site Code *	Group Size	Total Detections
AO	1.6 ± 1.1	152
AR	2.0 ± 1.7	91
DF	-	0
DR	-	0
KK	1.2 ± 0.7	87
KR	1.3 ± 0.6	16
SA	-	0
SV	1.1 ± 0.2	18
TR	1	6
XM	1	3

* Abbreviations of site codes: DR: Dromonas; SA: Selladi Arnion; TR: Tripilos; DF: Den of Fox; SV: Selladi Voullas; KK: Kokinokremmos; AO: Apo Orkontas; AR: Artratsa; XM: Xeromoutas; KR: Koriftos.

**Table 6 animals-12-03060-t006:** Model averaged estimates and significance of the fixed effect variables predicting mouflon visitation rate at the 6 hr time scale, as measured in the number of independent mouflon detection events at ten monitored water troughs (R^2^ = 0.562; negative binomial generalized mixed-effect model with site (water trough) and day as nested random effects (fixed slope, random intercept)).

Variables	Estimate	SE *	z-Value	Pr(>|z|) **
intercept (β_ο_)	−6.726	2.219	2.089	0.037
distance to roads (log)	−1.384	0.451	3.067	0.002
temperature (24 h max)	0.488	0.063	7.806	<0.0001
time of day—evening	0.382	0.173	2.205	0.027
time of day—late morning	0.712	0.163	4.355	<0.0001
time of day—midday	0.644	0.165	3.893	<0.0001
precipitation (past 7 days)	0.015	0.012	1.187	0.235
distance to terrace/cliff (log)	−0.117	0.245	0.476	0.634
precipitation (24 h)	−0.017	0.042	0.403	0.687

* SE: Standard Error; ** The Pr(>|z|) corresponds to the *p*-Value associated with the z-Value column.

**Table 7 animals-12-03060-t007:** Model averaged estimates and significance of the fixed effect variables predicting mouflon visitation rate at the weekly time scale, as measured in the number of independent mouflon detection events at ten monitored water troughs (R^2^ = 0.979; negative binomial generalized mixed-effect model with site (water trough) and week as nested random effects (fixed slope, random intercept)).

Variables	Estimate	SE *	z-Value	Pr(>|z|) **
intercept (β_ο_)	−6.07	4.912	1.226	0.220
distance to roads (log)	−1.729	0.592	2.899	0.004
distance to terrace/cliff (log)	−0.455	0.334	1.350	0.177
temperature (mean 24 h max)	0.694	0.139	4.939	<0.0001
predator encounter rate	1.237	0.734	1.67	0.095
southern exposure	5.018	3.032	1.643	0.1

*** SE: Standard Error; ** The Pr(>|z|) corresponds to the *p*-Value associated with the z-Value column.

## Data Availability

Data reported in this study are contained within the article. The underlying raw data are available on request from the corresponding author.

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
