# Peer review of "Artificial Water Troughs Use by the Mountain Ungulate *Ovis gmelini ophion* (Cyprus Mouflon) at Pafos Forest"

_animals, 2022, doi:10.3390/ani12213060_

Round 1
Reviewer 1 Report
I have enjoyed reading this paper and learning about the Cyprus mouflon. However, I do have some concerns about the methodology, statistical analysis and interpretation, which I think should be addressed before the paper is published. The discussion also needs to be better structured. Please see my comments in the pdf attached.

Author Response
Dear Associate Editor,
The authors thank the associate editor and the reviewers for giving a positive assessment for the manuscript. Based on the comments and suggestions by the reviewers, the authors team managed the weak points that were mentioned. Surely, the reviewers’ comments are of great help for this revision and were used for improving the manuscript.

Reviewer 2 Report
This manuscript has a real practical significance, the studies are important. I really enjoyed to read this paper, I felt the structure of the paper to be good, easily followable, and it is clear that the authors tried to provide many different aspects of the topic, both with the analyses and in the discussion. The Authors used adequate field techniques, study design (with some weak points but considering that it is a field work in harsh conditions and not a lab), and performed suitable statistical analyses.
However, I feel that the manuscript has several weaknesses in this present form, which is provided in my comments in the manuscript. After making a major revision of the the ms I think it will be a valuable publication for the Readers.

Author Response

(The authors gave the same response as above.)

Round 2
Reviewer 1 Report
Thank you for addressing most of my comments.
I still feel that you should acknowledge somewhere in your manuscript the potential issue of all animals observed near the water troughs assumed to be there to drink water. Could you include in the methods a statement like the one you used to address my concern? Something like: it was not possible to establish with certainty if mouflons were drinking or only investigating the water stations as we did not have continuous video recording. Nevertheless, we used a mean visit time of >4 min to ensure that mouflons were actually visiting the water station and not just passing by.
Line 572: p<0.1? I am confused. This is NOT significant so you cannot say that the result is 'significant at p<0.1 level'. Please change this as it does not make sense.
Author Response
The author team thanks both of the reviewers for the valuable comments. The additional minor but specific comments by the reviewer #1 (2nd phase revision) were addressed in the current version. Both of the comments were adopted and the sentences edited as suggested by the reviewer.
For the first point the first paragraph of “2.2. Data Analysis” was modified as (with the added sentence underlined):
For each photo/video, the species, date, time, and site were recorded. Photos/videos of the same species with up to 30 minute intervals were considered as part of the same independent detection (henceforth referred to as detection) [45]. Only the time of the first photo/video was used in the analysis from a series of records. All animals observed next to the artificial water troughs were assumed to be there to drink water, regardless if they were actually drinking in the photos/videos or not, as there was not continuous video to establish with certainty if animals were drinking. Nevertheless, the mean visit duration was >4 min, suggesting that the mouflon were visiting the trough rather than just passing by. In the case of mouflon, the minimum herd size (i.e. max number of individuals recorded in a single photo/video) and the estimated age (adult, sub-adult, juvenile) and sex of observed individuals were recorded, even though it was not assumed that all animals of a herd were observed.
For the second point, the paragraph was modified as follows (with the rewritten sentences underlined):
At the weekly scale mouflons also visited more often water troughs located in areas with higher southern exposure (percent of area within 1 km radius facing SE to SW), but this observation was not significant (p=0.1). South facing slopes in the northern hemisphere are more exposed to solar radiation [77], and therefore the thermoregulation needs of mouflon foraging in such areas will be higher compared to areas with low southern exposure. The above two findings suggest that a possible relation of water trough use and its location in terms of southern exposure and vicinity to terraces/cliffs should be examined more closely in the future, as it could provide guidance on where additional water troughs could be placed.
Best regards,
Nicolas-George Eliades
(on behalf of the authors’ team)
Reviewer 2 Report
I have reviewed the improved version of the manuscript. I found that the Authors adequately considered the comments and greatly improved the quality of the manuscript. I recommend this manuscript for publication.
Author Response

(The authors gave the same response as above.)
